# Transoral Outlet Reduction (TORe) for the Treatment of Weight Regain and Dumping Syndrome after Roux-en-Y Gastric Bypass

**DOI:** 10.3390/medicina59010125

**Published:** 2023-01-08

**Authors:** Landry Hakiza, Adrian Sartoretto, Konstantin Burgmann, Vivek Kumbhari, Christoph Matter, Frank Seibold, Dominic Staudenmann

**Affiliations:** 1Gastroenterology Service Intesto, 3012 Bern, Switzerland; 2Gastroenterology Service Intesto, Hôpital Fribourgeois, University of Fribourg, 1700 Fribourg, Switzerland; 3The BMI Clinic, Double Bay, NSW 2028, Australia; 4Division of Gastroenterology and Hepatology, Mayo Clinic, Jacksonville, FL 32224, USA

**Keywords:** endoscopic transoral outlet reduction, bariatric endoscopy, obesity, gastric bypass, dumping syndrome, weight regain

## Abstract

Obesity is a chronic relapsing disease of global pandemic proportions. In this context, an increasing number of patients are undergoing bariatric surgery, which is considered the most effective weight loss treatment for long-term improvement in obesity-related comorbidities. One of the most popular bariatric surgeries is the Roux-en-Y gastric bypass (RYGB). Despite its proven short- and long-term efficacy, progressive weight regain and dumping symptoms remain a challenge. Revisional bariatric surgery is indicated when dietary and lifestyle modification, pharmaceutical agents and/or psychological therapy fail to arrest weight regain or control dumping. However, these re-interventions present greater technical difficulty and are accompanied by an increased risk of peri- and postoperative complications with substantial morbidity and mortality. The endoscopic approach to gastrojejunal anastomotic revision, transoral outlet reduction (TORe), is used as a minimally invasive treatment that aims to reduce the diameter of the gastrojejunal anastomosis, delaying gastric emptying and increasing satiety. With substantial published data supporting its use, TORe is an effective and safe bariatric endoscopic technique for addressing weight regain and dumping syndrome after RYGB.

## 1. Introduction

Obesity is currently one of the greatest public health challenges, with substantial economic implications. According to estimates published by the World Health Organization (WHO) on 4 March 2022, more than one billion people worldwide suffer with obesity—650 million adults, 340 million adolescents and 39 million children [1]. The WHO estimates that by 2025, approximately 167 million people (adults and children) will become less healthy due to overweight or obesity [1].

The measures implemented against obesity usually include conservative treatment, such as lifestyle modification and drug therapy, as well as bariatric surgery; however, in most patients, sustained weight loss is not achieved [2,3,4].

Pharmacologic treatments allow an average loss of 10% of body weight (BW), and newer drugs under investigation seem even more promising [5]. However, weight regain is universally observed upon discontinuation of treatment [6].

Contemporary guidelines on the management of morbid obesity recognize bariatric surgery (BS) as the gold standard for weight loss and the improvement of obesity-related comorbidities [7,8,9]. Patients with obesity are generally considered eligible for BS at a BMI greater than 40 kg/m^2^, or greater than 35 kg/m^2^ when accompanied by serious weight-related comorbidities, such as type 2 diabetes mellitus, (T2DM) hypertension or obstructive sleep apnea [9]. 

The two most commonly performed bariatric procedures worldwide are the Roux-en-Y gastric bypass (RYGB) (39.6%) and the sleeve gastrectomy (SG) (45.9%) [9]. Compared to SG, RYGB confers superior clinical efficacy in terms of weight loss and in the remission of comorbidities, particularly T2DM [7,9,10].

## 2. Methods

A comprehensive search of several English-language databases and conference proceedings from 1990 to 2022 was conducted. The databases included PubMed, MEDLINE, EMBASE, Web of Science databases, Google Scholar and SCOPUS, with PubMed being the main database used. Secondary weight regain after gastric bypass surgery, dumping syndrome after gastric bypass, transoral outlet reduction and endoscopic suturing were used as keywords. Literature screening was independently performed by two authors (L.H. and D.S.), with research focusing on studies with long-term outcomes (from 2 to 5 years post-TORe).

## 3. TORe for Weight Regain after RYGB

Weight recidivism is a common complication following RYGB surgery. On average, patients regain between 20 and 30 % of lost weight, and moreover, excessive weight gain is experienced by over one third of patients [11,12]. Weight regain after gastric bypass is often multifactorial and can be attributed to eating patterns, and psychological and social factors. However, dilatation or enlargement of the gastrojejunal anastomosis of >30 mm is a significant predictor of weight regain following RYGB [13,14,15]. Due to the technical complexity of the anatomy, surgical re-intervention is accompanied by a high risk of complications and an increase in postoperative morbidity and mortality [16]. As an alternative, TORe was developed in 2013 as an endoscopic procedure focusing on reducing the size of the gastrojejunal anastomosis (GJA) [17]. The first interventional study included 25 patients with an average weight regain of 24 kg after RYGB [17]. This study described endoscopically reducing the diameter of the anastomosis by an average of 77.3% which was associated with an average weight loss of 11.5 kg, 11.7 kg and 10.8 kg at 3, 6 and 12 months, respectively [17]. 

Vargas et al. demonstrated in a multicenter study that TORe is a safe, reproducible and effective approach to managing weight recidivism after RYGB [18]. The average weight loss at 6, 12 and 18 months was 9.31 ± 6.7 kg, 7.75 ± 8.4 kg and 8 ± 8.8 kg, respectively, and no serious adverse events were reported [18].

Recently, a five-year outcome study concluded that endoscopic revision of the GJA for weight regain is a durable approach [12]. Total body weight loss (TBWL) of 8.5% at 1 year (n = 276/331 patients), 6.9% at 3 years (n = 211/331), and 8.8% at 5 years after TORe was shown [12]. In addition, the majority of the patients (77%) experienced complete cessation of weight gain and 62% were able to maintain a TBWL of >5% at 5 years [12].

Furthermore, an American group assessed patients’ ability to lose weight after TORe and the magnitude of the reduction of the GJA [19]. They demonstrated that patients who had a larger reduction in diameter had a more significant TBWL. After 3 and 5 years following TORe, TBWL was 5.3 ± 9.1 kg and 3.9 ± 13.1 kg, respectively [19].

## 4. TORe for Dumping Syndrome after RYGB 

Dumping syndrome (DS) is a postprandial phenomenon in which patients present with a constellation of gastrointestinal and vasomotor symptoms, including tachycardia, fatigue, syncope, and occasionally, shock and seizures due to profound hypoglycemia [20]. Symptoms may occur early (within 1 h of a meal) or up to 3 h later, the latter being associated with postprandial hypoglycemia. As the name suggests, DS occurs, in part, due to rapid gastric emptying, leading to rapid passage of food into the small intestine [21,22]. The patient’s typical history and blood sugar determination inform the diagnosis. The Sigstad score (a score >7 is strongly suggestive of dumping) and questionnaires may also be helpful [21].

A conservative stepwise approach is currently recommended, starting with dietary changes in the form of more frequent meals with increased protein content and lower overall carbohydrate content, favoring complex carbohydrates [23,24]. If dietary measures prove unsuccessful, drug therapy can be initiated with acarbose, calcium antagonists or GLP-1 analogues [25].

However, dietary restrictions and pharmacological treatments are often ineffective or poorly tolerated [21,22]. In these cases, TORe provides a solution by reducing the speed of gastric emptying, however there is no clear consensus in the literature regarding the place of surgical re-intervention in treating dumping syndrome [21,26,27,28]. 

A large study involving 115 patients from two large academic centers in the United States and Germany supported TORe as an effective and safe adjuvant therapy to lifestyle and pharmacologic treatment of refractory DS [29]. The Sigstad score reduced significantly after only 3 months post-TORe, with the mean sore changing from 17 ± 6.1 to 2.6 ± 1.9 [29]. Similarly, Brown et al. demonstrated a 90% rate of resolution of DS after only 3 months of revision [30].

A recent retrospective study was published in October 2022, where 83% of the patients had a long-term follow-up at a mean of 3.45 years [31]. This retrospective study also found that the presence of gastro-esophageal reflux disease prior to TORe was a predictor of the resolution of DS following the procedure. While the difference was small, it achieved statistical significance (69% vs. 62%; *p* = 0.03) [31].

In this context, TORe is not only an effective approach to managing weight recidivism after RYGB, but also to treating DS.

## 5. TORe Technique

TORe is currently the most frequently used technique for the reduction of a dilated GJA (Figure 1A). The intervention is usually performed under general anesthesia. A double-lumen gastroscope is passed through a proprietary overtube of 25 cm in length, and CO_2_ is used for insufflation. It can be carried out on an outpatient basis, and it is typically performed with argon plasma coagulation (APC) combined with full-thickness suturing achieved using the OverStitchTM device (Apollo Endosurgery, Austin, TX, USA) [15,32,33]. This combined technique allows for greater durability of anastomotic reduction by inducing fibrosis of the GJA [34,35]. The first step of the procedure is to ablate the gastric rim of the anastomosis via APC (forced APC, 0.8 L/min with 30–70 watts) (Figure 1B), followed by a circumferential, transmural endoscopic suture (Figure 1C). Suturing is mainly performed via the creation of a purse-string, or alternatively, by placing interrupted sutures at the GJA [29]. The purse-string technique is, however, generally favored, as it results in more significant weight loss at one year than interrupted suture patterns [36]. Ideally, a dilation balloon (CRE balloon dilator, Boston Scientific, Marlborough, MA, USA) is introduced through the second channel of the endoscope and inflated to a diameter of 8–10 mm (Figure 1D) to size the GJA before the suture is tightened and cinched over the balloon, allowing the GJA to be precisely sized (Figure 1E and Appendix A).

There are several other TORe techniques described in the literature [35,37,38,39,40]. Initially, some studies demonstrated efficacy using APC alone in the GJA, which was relatively simple to perform, and even feasible with patients under conscious sedation [41,42,43,44]. Jaruvongvanich et al. reported a meta-analysis showing that both full-thickness suturing plus APC (ft-TORe) and argon plasma mucosal coagulation alone (APMC-TORe) offer comparable weight loss outcomes and safety profiles, but the AMPC-TORe technique usually requires multiple endoscopic sessions [35].

Barola et al. performed a two-fold running suture TORe technique with a significant reduction in BMI (5.5 + 5.0%, *p* < 0.001 at mean follow-up of 113.2 ± 75.7 days (15.4%)); however, 15.4% of the patients developed a gastric stenosis that was treated with balloon dilation [39].

A new approach combining the restriction component of TORe followed by type 1 surgical distalization of the Roux limb may be another alternative for managing weight regain in high-BMI patients after RYGB; however, this could result in greater malabsorption, leading to greater deficiency syndrome [40].

More recently, we have seen the emergence of a novel, modified technique: first performing an endoscopic submucosal dissection (ESD) before applying endoscopic sutures. This is known as ESD-TORe [37,38]. A retrospective study compared patients who underwent modified ESD-TORe vs. APC -TORe. At 12 months, the ESD-TORe group experienced greater weight loss compared with the traditional TORe group (12.1% ± 9.3% vs. 7.5% ± 3.3% TBWL) [38]. However, this technique resulted in a higher rate of major complications (21.1% for ESD-TORe vs. 8.77% for APC-TORe) which, combined with the technical difficulty of ESD, limits its widespread adoption [34,38].

On the other hand, the TORe procedure has demonstrated a high degree of safety, with only minor intraprocedural adverse effects (AE) such as superficial lacerations of the esophageal mucosa due to the use of the overtube [12,17,18,19,29,30,31,32,33,45,46,47]. Additional postprocedural serious AEs include bleeding from marginal ulceration and GJA stenosis [20,21,22,25,30,31,32,33,34,40,41,42]. In general, AEs can be successfully managed endoscopically without the need for surgery. 

## 6. Discussion

Despite the efficacy and durability of RYGB, weight regain and the return of comorbid conditions, as well as DS, is of major concern [15,18,29,31,45,46,48]. The underlying causes are multifactorial, and therefore, its management requires a multidisciplinary approach, in collaboration with general practitioners, surgeons, dietitians, endocrinologists, psychiatrists or psychologists and gastroenterologists [49,50]. One of the most common factors contributing to weight regain and DS after RYGB is a dilated GJA [15,18,19,32,33,46]. Initially, this was treated with revisional bariatric surgery such as pyloric reconstruction, the conversion of Billroth II to Billroth I anastomoses, jejunal interposition and Roux-en-Y conversion [51]. However, the surgical approach is associated with increased risk and limited effectiveness [21,26,27,28]. 

The TORe technique has now repeatedly demonstrated its efficacy, safety and favorable long term results for up to 5 years in the management of weight regain after RYGB [12,13,15,19,32,38]. Patients are able to maintain a TBWL of 12.5% at 5 years [19,38,45]. Recent studies have also shown that it can be used as a minimally invasive treatment for refractory DS, demonstrating an 80% and 84% resolution of DS at 2- and 3.5-year follow-ups, respectively [30,31]. Moreover, it has been illustrated that TORe is accompanied by a very low risk of serious adverse events, and no deaths have been causally associated with the procedure [12,19,31,52]. Given the very low risk of severe complications, TORe is easily repeatable if necessary [29,45]. This growing body of evidence supports the role of TORe as an emerging standard of care in the treatment of weight regain and DS in patients with prior RYGB, now superseding surgical intervention.

## 7. Conclusions

TORe represents an endoscopic bariatric technique that has been proven to be safe and durably efficacious in managing weight regain, as well as DS, post-RYGB. While first-line treatment for these conditions remains lifestyle and pharmacologic therapies delivered in a multidisciplinary setting, TORe has effectively replaced revisional surgery as a first-line interventional therapy owing to its superior safety profile, lower resource requirement and demonstrated clinically meaningful efficacy.

## Figures and Tables

**Figure 1 medicina-59-00125-f001:**
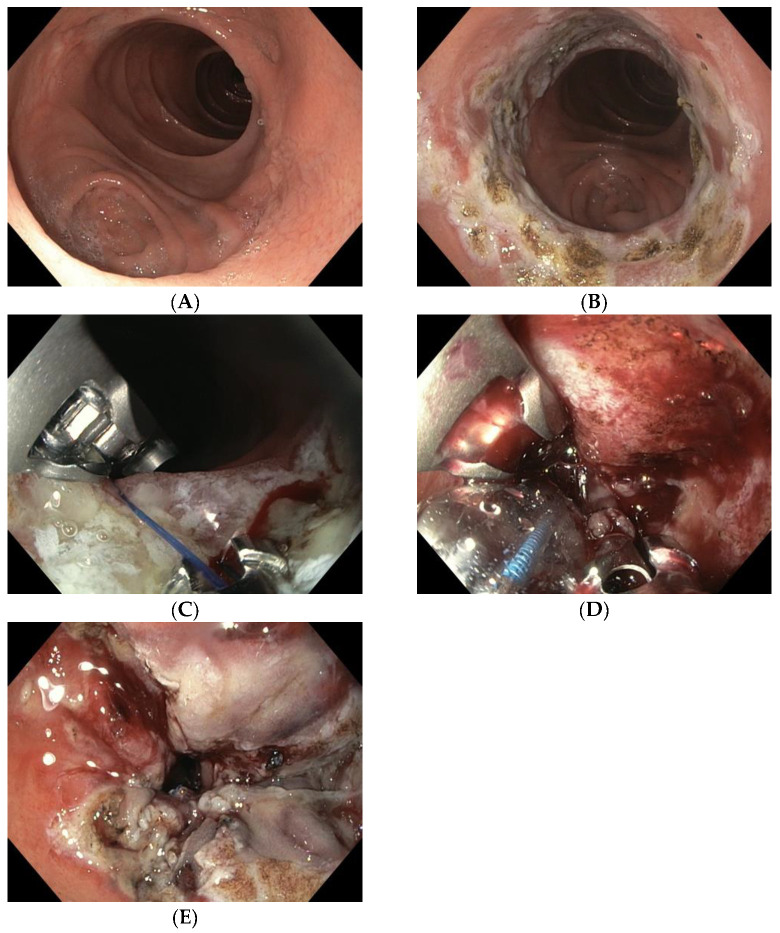
(**A**) Dilated GJA; (**B**) ablation of the gastric rim via APC; (**C**) suturing the anastomosis with the Apollo Overstitch system; (**D**) suture size control using an 8 mm CRE balloon; (**E**) narrowed GJA after TORe.

## Data Availability

Not applicable.

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
