# Peer review of "Transoral Outlet Reduction (TORe) for the Treatment of Weight Regain and Dumping Syndrome after Roux-en-Y Gastric Bypass"

_medicina, 2023, doi:10.3390/medicina59010125_

Round 1

Reviewer 1 Report

Dear Authors, 

This is a review article, perhaps you should include a graph showing where and how you selected the articles you are presenting. 

Otherwise, you have gone into the aspects of TORe (weight gain, dumping syndrome) and have worked out the advantages and possible disadvantages well. 

As it is imperative that this technique becomes known and integrated into the standard treatment algorithm for both weight gain and dumping syndrome, I recommend publication after minor improvements. 

What I do not agree with is your statement about combining TORe with distalisation of the bypass. I do not think this is advisable, as such a combination of therapies (Increase of restriction and significant increase of malabsorption)  must in any case lead to deficiency symptoms. 

Author Response

Point 1: This is a review article, perhaps you should include a graph showing where and how you selected the articles you are presenting. Response 1: We would like to thank the editors and reviewers for their excellent comments and feel that the paper is now stronger as a result. We have addressed all the questions raised to the best of our abilities below. Indeed, we based our review article on the main studies evaluating traditional full-thickness TORe (APC-Apollo Overstitch) for weight regain and dumping syndrome after gastric bypass. These articles were chosen on the basis of their ability to highlight the long-term clinical evolution (from 3 months to 5 years), the reproducibility of the TORe technique, the large number of patients participating in the studies, and the ease of verification of the statistical data from each study. Instead of adding a graphic we propose to add the following paragraph regarding the search strategy: “A comprehensive search of several English-language databases and conference proceedings from 1990 to 2022 was conducted. The databases included PubMed, MEDLINE, EMBASE, Web of Science databases, Google-Scholar and SCOPUS, with PubMed being the main database used. Secondary weight regain after gastric bypass surgery, dumping syndrome after gastric bypass, transoral outlet reduction and endoscopic suturing were used as keywords. Literature screening was independently performed by two authors (LH and DS), with research focused on studies with long-term outcomes (from 2 to 5 years post-TORe).” Point 2: What I do not agree with is your statement about combining TORe with distalisation of the bypass. I do not think this is advisable, as such a combination of therapies (Increase of restriction and significant increase of malabsorption) must in any case lead to deficiency symptoms . Response 2: Thank you for pointing this out. You are certainly right, it is indeed a new method but as you point out it will probably increase vitamin deficiencies and deficiency syndrome. This has now been corrected : A new approach combining the restriction component of TORe followed by type 1 surgical distalization of the Roux limb may be another alternative for managing weight regain in high BMI patients after RYGB, however this could result in greater malabsorption leading to greater deficiency syndrome.

Reviewer 2 Report

I would like to thank the editors for inviting me to review this work. This is a very interesting paper on Transoral Outlet Reduction (TORe) for the treatment of weight regain and dumping syndrome after Roux-en-Y gastric bypass. I believe that this study will be useful in clinical practice, however a few changes need to be made.

Please revise the validity of the cited works again. Eg 9 - don't the authors think this should be a guideline? or 10 - the source should be the original current paper on bariatrics  around the world (see: https://doi.org/10.1007/s11695-018-3593-1).

I don't see the material and methods of this review. I consider it essential in a scientific publication.

In my humble opinion, a description of the procedure should precede further analysis. 

Author Response

Point 1: Please revise the validity of the cited works again. Eg 9 - don't the authors think this should be a guideline? or 10 - the source should be the original current paper on bariatrics around the world (see: https://doi.org/10.1007/s11695-018-3593-1). Response 1: Thank you for the critical review and feedbacks of our review article: “Transoral Outlet Reduction (Tore) for the Treatment of Weight Regain and Dumping Syndrome After Roux-En-Y Gastric Bypass “. We thoroughly appreciated your comments and suggestion. We have carefully reviewed the validity of the cited worked and made the necessary changes in this respect. Thank you again for your attention. Point 2: I don't see the material and methods of this review. I consider it essential in a scientific publication. In my humble opinion, a description of the procedure should precede further analysis. Response 2: We thank you for pointing out this omission. We have now added the following methods of this review: A comprehensive search of several English-language databases and conference proceedings from 1990 to 2022 was conducted. The databases included PubMed, MEDLINE, EMBASE, Web of Science databases, Google-Scholar and SCOPUS, with PubMed being the main database used. Secondary weight regain after gastric bypass surgery, dumping syndrome after gastric bypass, transoral outlet reduction and endoscopic suturing were used as keywords. Literature screening was independently performed by two authors (LH and DS), with research focused on studies with long-term outcomes (from 2 to 5 years post-TORe).

Round 2

Reviewer 2 Report

Thank you for considering my opinions. I think the authors should add flow chart of the study. We do not know how many papers have been qualified for review, how many papers have been reviewed. This will increase the value of the work.

Author Response

Point 3: I think the authors should add flow chart of the study. We do not know how many papers have been qualified for review, how many papers have been reviewed. This will increase the value of the work Response 3: Thanks for your comment. Of course, a flow chart of the studies would have been more desirable and could increase the value of the work. Unfortunately, we did not carry out our review according to the Preferred Reporting Items for Systematic Reviews and Meta-analyses (PRISMA) assessment and doing this retrospectively does not seem reliable to us. Moreover, a well reported flow chart is helpful for readers to understand the sequence of the review process and line out any source of bias, however Vu-Ngoc et al. also showed mayor concerns about the quality and reporting components in the flow diagram [1].The analysis also highlights that although flow diagram utilization was correlated with the methodological quality of systematic review, it was not associated with the impact factor of journals where they are published. It would be interesting in the future to evaluate the impact and quality improvement of a good flow chart on the impact factor in general [ 2]. In addition, most published reviews on this topic do not have a flow chart [3–7]. We recognize that we need to improve our efforts to help readers gain a comprehensive understanding of the review process and therefore added the paragraph regarding methods. References 1. Vu-Ngoc, H., Elawady, S. S., Mehyar, G. M., Abdelhamid, A. H., Mattar, O. M., Halhouli, O., Vuong, N. L., Ali, C. D. M., Hassan, U. H., Kien, N. D., Hirayama, K., & Huy, N. T. (2018). Quality of flow diagram in systematic review and/or meta-analysis. PLOS ONE, 13(6), e0195955. https://doi.org/10.1371/journal.pone.0195955 2. Stovold, E., Beecher, D., Foxlee, R., & Noel-Storr, A. (2014). Study flow diagrams in Cochrane systematic review updates: an adapted PRISMA flow diagram. Systematic Reviews, 3(1), 54. https://doi.org/10.1186/2046-4053-3-54 3. Jirapinyo, P., Kröner, P., & Thompson, C. (2018). Purse-string transoral outlet reduction (TORe) is effective at inducing weight loss and improvement in metabolic comorbidities after Roux-en-Y gastric bypass. Endoscopy, 50(04), 371–377. https://doi.org/10.1055/s-0043-122380 4. Jirapinyo, P., Kumar, N., AlSamman, M. A., & Thompson, C. C. (2020). Five-year outcomes of transoral outlet reduction for the treatment of weight regain after Roux-en-Y gastric bypass. Gastrointestinal Endoscopy, 91(5), 1067–1073. https://doi.org/10.1016/j.gie.2019.11.044 5. Thompson, C. C., Chand, B., Chen, Y. K., DeMarco, D. C., Miller, L., Schweitzer, M., Rothstein, R. I., Lautz, D. B., Slattery, J., Ryan, M. B., Brethauer, S., Schauer, P., Mitchell, M. C., Starpoli, A., Haber, G. B., Catalano, M. F., Edmundowicz, S., Fagnant, A. M., Kaplan, L. M., & Roslin, M. S. (2013). Endoscopic Suturing for Transoral Outlet Reduction Increases Weight Loss After Roux-en-Y Gastric Bypass Surgery. Gastroenterology, 145(1), 129-137.e3. https://doi.org/10.1053/j.gastro.2013.04.002 6. Tsai, C., Steffen, R., Kessler, U., Merki, H., & Zehetner, J. (2019). Endoscopic Gastrojejunal Revisions Following Gastric Bypass: Lessons Learned in More Than 100 Consecutive Patients. Journal of Gastrointestinal Surgery, 23(1), 58–66. https://doi.org/10.1007/s11605-018-3961-0 7. Vargas, E. J., Abu Dayyeh, B. K., Storm, A. C., Bazerbachi, F., Matar, R., Vella, A., Kellogg, T., & Stier, C. (2020). Endoscopic management of dumping syndrome after Roux-en-Y gastric bypass: a large international series and proposed management strategy. Gastrointestinal Endoscopy, 92(1), 91–96. https://doi.org/10.1016/j.gie.2020.02.029
